# Genetic Diversity of the Only Natural Population of *Corylus avellana* L. in Kazakhstan and Prospects for Its In Vitro Conservation

**DOI:** 10.3390/biology14111472

**Published:** 2025-10-23

**Authors:** Svetlana V. Kushnarenko, Madina Omasheva, Natalya Romadanova, Moldir Aralbayeva, Nazgul Rymkhanova, Ulzhan Manapkanova, Roberto Botta, Paola Ruffa, Nadia Valentini, Daniela Torello Marinoni

**Affiliations:** 1Institute of Plant Biology and Biotechnology, 45 Timiryazev St., Almaty 050040, Kazakhstan; omasheva.madina@gmail.com (M.O.); nataromadanova@gmail.com (N.R.); moldiraralbayeva777@gmail.com (M.A.); n.rymkhanova@gmail.com (N.R.); ylzhanchikm@gmail.com (U.M.); 2Department of Agricultural, Forestry and Food Sciences, University of Turin, 10095 Grugliasco, TO, Italy; roberto.botta@unito.it (R.B.); paola.ruffa@unito.it (P.R.); nadia.valentini@unito.it (N.V.); daniela.marinoni@unito.it (D.T.M.)

**Keywords:** *Corylus avellana* L., genetic diversity, microsatellite markers, population structure, micropropagation, in vitro culture

## Abstract

**Simple Summary:**

*Corylus avellana* L., known in Europe as the common hazel, is extremely rare in Kazakhstan and listed as an endangered species. In this study, morphological characterization of the trees and leaves was carried out, as well as an analysis of their genetic background, to understand the population’s status and conservation prospects. The findings revealed that this group of shrubs has a high level of diversity, which will be highly valuable for future breeding programs. At the same time, the Kazakhstan population differs significantly from cultivated varieties. During the analysis, it was observed that the population is declining and not producing enough viable seeds. Therefore, young shoots were used to preserve the plants in laboratory conditions. This approach enabled the creation of a healthy collection that can be used to restore and protect the species. Our results highlight the need for urgent measures to conserve hazelnut in Kazakhstan and show that modern technologies can play a key role in preserving rare plants for future generations.

**Abstract:**

*Corylus avellana* L. is a rare and endangered species in Kazakhstan, included in the national Red Book. The results of morphological and genetic characterization of the sole known natural population of *C. avellana* in the Western Kazakhstan region are presented in this study. Sixty wild accessions were evaluated based on tree and leaf morphological traits using standard descriptors in accordance with Bioversity International guidelines. Genetic diversity was assessed using ten nuclear simple sequence repeat (SSR) markers. A total of 120 alleles were detected across the nuclear loci, with the number of alleles per locus ranging from 9 to 16 and an average of 12. The mean effective number of alleles (Ne) per locus was 3.862. A high level of intraspecific polymorphism was observed, with an average observed heterozygosity (Ho) of 0.70. The population showed considerable genetic diversity, as highlighted by a mean Shannon’s diversity index of 1.526. STRUCTURE, PCoA, and phylogenetic analyses confirmed strong differentiation between the wild Kazakh population and the cultivated hazelnut germplasm. Due to the lack of viable seeds, in vitro conservation was initiated using vegetative shoots. A two-step disinfection protocol, involving Plant Preservative Mixture and mercuric chloride, significantly improved explant survival, enabling successful establishment of an aseptic in vitro collection. These findings highlight the urgent need for targeted conservation strategies and show the potential of biotechnological approaches for safeguarding Kazakhstan’s only natural *C. avellana* population.

## 1. Introduction

*Corylus avellana* L. (European hazelnut) is a widespread deciduous shrub native to the temperate zone of Europe and Western Asia. It exhibits high ecological plasticity and holds considerable ecological and economic importance. In natural ecosystems, *C. avellana* serve as a forage plant and melliferous species, and contributes to forest formation processes. Economically, it is a key species in the global nut industry. Based on genome-wide data comprising 4894 single-nucleotide polymorphisms (SNPs), the most recent common ancestor of the genus *Corylus* is estimated to have originated approximately 36 million years ago in the region corresponding to present-day southwestern China [1]. Subsequent climate-driven migrations during the Cenozoic era facilitated the dispersal of various *Corylus* species across Europe, North America, and Central Asia, with *C. avellana* representing a lineage that colonized Europe, the Caucasus, and Asia Minor [2].

The current distribution of *C. avellana* spans much of Europe—from the British Isles and Scandinavia to the Balkans and the Caucasus—as well as parts of western Asia Minor and Iran. In Kazakhstan, a single known population of *C. avellana* has been recorded in the Western Kazakhstan region, within the floodplain of the Zhaiyk (Ural) River. This relict population occupies a small area of approximately 0.2 hectares and is listed in the Red Book of Kazakhstan as a very rare and endangered species [3,4,5]. A population assessment conducted in 2018–2019 revealed an alarmingly poor conservation status, despite the site being nominally protected as the “Dubrava” State Botanical Reserve. Approximately 70% of individuals failed to produce nuts, showed high levels of pest infestation, and exhibited minimal natural regeneration [6]. These findings emphasize the complex biogeographic history of *C. avellana* and reinforce the urgent need to conserve local populations, especially those at the species’ range margins, with the aim of preserving unique genetic traits.

In situ conservation alone is insufficient to prevent the irreversible loss of such valuable genetic resources. Modern biotechnological approaches, particularly micropropagation, are increasingly recognized as essential tools for the conservation, multiplication, and restoration of rare plant populations. Developing an efficient in vitro propagation protocol supports the establishment of in vitro collections and ensures the long-term preservation of genetic lineages. Furthermore, it enables the production of healthy plant material for the creation of ex situ mother plant collections, which can support future reintroduction and population reinforcement efforts.

The micropropagation of hazelnut has been extensively investigated worldwide. Classic protocols [7,8] typically utilize Driver and Kuniyuki Walnut (DKW) or Woody Plant Medium (WPM), supplemented with cytokinins such as 6-benzylaminopurine (BAP) or meta-topolin, and auxins such as indole-3-butyric acid (IBA) for rooting. Recent research has focused on improving multiplication rates [9], reducing hyperhydricity [10], and maintaining genetic fidelity [11]. Additives such as glucose [12], iron chelates (Fe-EDDHA), and antioxidants have been shown to improve explant viability and culture stability. However, most existing protocols have been developed for European cultivars or hybrids and require adaptation for local genotypes, particularly wild populations, which may harbor unique resistance traits and greater adaptive potential.

Assessing the genetic diversity of *C. avellana* is essential for the conservation of valuable alleles and for understanding responses to biotic and abiotic factors, while its potential contribution to breeding programs remains to be further assessed. Over recent decades, a wide range of molecular markers have been employed, including microsatellites (SSR) [13,14,15,16,17,18], ISSR and AFLP [19], RAPD [14,20], and next-generation approaches such as SNP genotyping [21] and genotyping-by-sequencing (ddRAD-seq) [22,23]. These studies consistently show that the highest levels of genetic diversity are found in the Black Sea region (particularly Turkey), as well as in Southern and Southeastern Europe—especially Italy, Spain, Slovenia, and Romania. Both wild and cultivated populations display high levels of heterozygosity [17,19,21].

The aim of the present study was to assess the morphological and genetic diversity of the relict *C. avellana* population in Kazakhstan using SSR markers and to initiate the first in vitro culture of this population. To the best of our knowledge, this represents the first molecular genetic analysis of *C. avellana* in Kazakhstan and the first successful establishment of an in vitro collection of this species from a native population.

## 2. Materials and Methods

### 2.1. Plant Material and Collection Site

The object of this study was a *Corylus avellana* L. population located in western Kazakhstan, approximately 50 km northeast of the city of Oral, on the left bank of the Zhaiyk (Ural) River. Geographic coordinates were recorded using an eTREX^®^H Garmin GPS device (eTREX^®^ H, Garmin Ltd., Olathe, KS, USA). Latitude, longitude, and altitude above sea level were determined, and vegetation at the collection site was described.

### 2.2. Data Collection and Phenotypic Evaluation

Expeditionary field trips were organized in September 2023 and May 2024. A phenotypic description of 60 *C. avellana* accessions (shrubs) was carried out using descriptors developed by Bioversity International [24]. One-year-old shoots, 30–40 cm long, were collected in December 2023 for in vitro culture initiation. Leaves from the same 60 accessions were sampled for DNA extraction and molecular genetic analysis.

### 2.3. Genomic DNA Extraction and Microsatellite Genotyping

Leaves from 60 wild hazelnut accessions collected during the expedition were dried using silica gel. Approximately 100–120 g of fresh leaves was placed in zip-lock plastic bags between two layers of filter paper containing 1 g of pre-prepared silica gel (Sigma-Aldrich, St. Louis, MO, USA). The silica gel had been pre-conditioned in a drying oven at 160 °C for 6 h. The leaves were stored in the silica gel bags during transport for 3–5 days, after which a portion was used for DNA extraction, while the remaining material was preserved at −80 °C for long-term storage. Genomic DNA was extracted from 80 mg of dried leaf tissue following the method described by Edwards et al. [25]. DNA quality and quantity were assessed by 1% agarose gel electrophoresis and quantified using a NanoDrop 1000 spectrophotometer (Thermo Scientific, Waltham, MA, USA). DNA was then diluted to a working concentration of 10 ng/µL. Ten nuclear SSR microsatellites were used for genotyping the 60 wild samples: CaT-B107, CaT-B501, CaT-B502, CaT-B503, CaT-B504, CaT-B505, CaT-B507, CaT-B508 [26], Cac-BO20, and Cac-B028 [27]. SSR analysis was performed according to Boccacci et al. [28]. Amplified products for all 10 SSRs were analyzed using a 3130 Genetic Analyzer (Applied Biosystems, Waltham, MA, USA).

### 2.4. Genetic Diversity Analysis

TANDEM v1.09 software was used to calculate the conversion factor for each SSR marker [29]. Genetic diversity parameters including the numbers of different alleles (Na) and their frequency, the number of effective alleles (Ne), the number of private alleles (No), expected and observed heterozygosity (He, Ho), the Shannon–Weaver Diversity Index (I), and Nei’s genetic distance were estimated using GenAlEx 6.5 [30,31]. Polymorphism information content (PIC) was calculated according to Botstein et al. [32] using the Excel Microsatellite Toolkit, version 3.1.1. Allelic richness (Rs) and private allele richness (PR) were calculated by using the rarefaction method via HP-RARE software, version 1.1 [33].

### 2.5. Population Structure Analysis

To analyze the genetic structure of the 60 wild *C*. *avellana* accessions from Kazakhstan and the 105 cultivars provided from the University of Turin, Italy, a Bayesian model-based approach was used as implemented in STRUCTURE 2.3.4. software [34]. The admixture model was applied as recommended by Hubisz et al. [35]. The burning period and Markov Chain Monte Carlo (MCMC) repetitions each consisted of 100,000 iterations. K values ranging from 2 to 5 were tested across five independent runs. The optimal number of clusters (K) was determined using the ΔK method by Evanno et al., implemented in STRUCTURE SELECTOR (https://lmme.qdio.ac.cn/StructureSelector, accessed on 15 July 2025) [36,37]. Individuals with a membership coefficient > 0.9 to the wild gene pool were considered pure wild types, enabling us to estimate the proportion of pure wild individuals in each population [38]. Principal coordinates analysis (PCoA) was performed to explore genetic relationships.

A phylogenetic tree was constructed using DARwin 5.0 software, employing the unweighted neighbor-joining method [39]. Cluster reliability was assessed through bootstrap analysis with 10,000 repetitions. The resulting tree was visualized and edited using Dendroscope and Interactive Tree of Life (iTOL), version 6.9 [40,41].

### 2.6. In Vitro Culture Initiation

One-year-old shoots, 30–40 cm long, were initially washed in soapy water, then in diluted (1:4) commercial bleach “Domestos” (5% sodium hypochlorite, 5% surfactant anions), followed by rinsing in tap water. The bases of cuttings were placed in distilled water at room temperature (24 ± 1 °C) for bud sprouting. Apical segments, 2–3 cm long, from expanding lateral buds were washed in soapy water and surface disinfected with 0.1% mercury chloride and 2–3 drops of “Tween 20” for 7 min, after which they were washed with sterile distilled water. As an additional treatment, 5% (*v*/*v*) PPM (Plant Cell Technology, Washington, DC, USA) solution with Hazelnut Medium (HM) basal salts [42] was used for shaking the explants for 30, 60, or 90 min. Following PPM treatment, the apices were washed three times in sterile distilled water and placed into 25 mm × 150 mm glass tubes on HM with 4.4 µM 6-benzylaminopurine (BAP), 0.04 µM indolyl-3-butyric acid (IBA), 4 g·L^−1^ agar (Plant TC agar, PhytoTechnology Laboratories^®^, Shawnee Mission, KS, USA), 1.75 g·L^−1^ Gerlite™ (PhytoTechnology Laboratories), and 20 g·L^−1^ glucose. The pH was adjusted to 5.5 with 0.1 N NaOH and autoclaved at 121 °C for 20 min. The number of viable shoots, bacterial and fungal contamination, and necrosis were recorded visually.

Two weeks after in vitro culture initiation, shoots were indexed for endophytic contamination using 523 detection medium [43]. Contaminated shoots were discarded, and further micropropagation was carried out only with indexed bacteria-free shoots. In vitro cultures were maintained at 24 °C, under 40 µmol·m^−2^·s^−1^ light intensity, with a 16 h photoperiod, and were subcultured onto fresh media every four weeks. The plant growing room is equipped with two types of OPPLE tubular fluorescent lamps: YK21RR 16/G 21 W 6500 K RGB and YK21RL 16/G 21 W 4000 K RGB (ElectroComplex Corporation, Almaty, Kazakhstan).

### 2.7. Statistical Analysis

In vitro experiments were carried out in triplicate, with 10–15 explants per treatment. The results are presented as the mean value ± standard error (M ± SE) and were analyzed using SYSTAT 13.0 software [44]. Differences were considered statistically significant at *p* < 0.05.

## 3. Results

### 3.1. Phenotypic Evaluation of Corylus avellana

A field survey of the *Corylus avellana* L. population in West Kazakhstan was conducted in September 2023 and May 2024. This sole *C*. *avellana* population is located approximately 50 km northeast of Oral (Uralsk), occupying around 0.2 hectares on a river terrace along the left bank of the Zhaiyk (Ural) River within the territory of the “Dubrava” State Botanical Reserve (Figure 1 and Appendix A).

A total of 60 individual shrubs were evaluated using the standard hazelnut descriptors provided by Bioversity International [24]. The elevation of the site ranges from 124 to 161 m above sea level. The dominant woody species in the habitat include the native *Quercus robur* L. and *Ulmus laevis* Pall., as well as the invasive *Acer negundo* L. Observations indicated that *C. avellana* plants in this population were generally in poor condition, forming dense cluster and regenerating exclusively through vegetative (clonal) shoot formation, while sexual reproduction via seeds was extremely limited. The population structure was heavily skewed toward mature individuals, with 83.3% of the accessions classified as adult shrubs. Only 6.7% were classified as juvenile plants, while 10.0% showed signs of senescence and decline (Figure 2A,B). Leaf morphology was predominantly ovate, although some rounded forms were also observed (Figure 2D,E). The average leaf length and width were 7.43 ± 0.92 cm and 5.58 ± 0.71 cm, respectively. In the surveyed population, pest damage was observed in 7% of shrubs, primarily caused by oak leafrollers (Tortricidae), red slugs (*Arion rufus*), and weevils (Curculionidae). Fungal diseases, such as leaf spot, were recorded on only a single specimen, indicating the absence of systemic infections and allowing the overall sanitary condition of the population to be assessed as satisfactory.

A previous assessment of this population in 2018 also reported poor overall condition, with only about 30% of plants producing isolated fruits [6]. However, by 2023, no fruiting was observed. During an assessment in May 2024, the site was found to be inundated due to spring flooding (Figure 2C), with many shrubs partially submerged and exhibiting further sign of deterioration.

Follow-up evaluations in both 2023 and 2024 confirmed the ongoing decline of this population. Fruiting remained completely absent, and a high incidence of pest and disease damage was recorded (Figure 2B,F). These observations underscore the critical conservation status of this relict population, which remains the only known natural occurrence of *C. avellana* within the territory of Kazakhstan.

### 3.2. Genetic Analysis

A set of ten simple sequence repeat (SSR) markers was used to evaluate the genetic diversity of the 60 wild *С. avellana* accessions. All 10 SSR loci were found to be polymorphic, and each accession exhibited a unique genotype, indicating the absence of duplicates and highlighting the distinct genetic diversity present within the population.

The 10 SSR loci amplified a total of 120 alleles across the 60 samples, with an average of 12 alleles per locus (Table 1). Among these, locus CaT-B107 was the most polymorphic, yielding 16 alleles, whereas CaT-B505 was the least polymorphic with 9 alleles. The total number of alleles (Na) was considerably higher than the effective number of alleles (Ne), suggesting that only a subset of alleles contributed substantially to the overall genetic diversity. The mean polymorphic information content (PIC) across all loci was 0.70, confirming the high informativeness of the selected markers. The highest PIC value (0.81) was recorded at locus CaT-B504, identifying it as the most informative. Additional diversity parameters support these findings: the probability of identity (PI) ranged from 0.05 to 0.26, and the Shannon–Weaver Diversity Index (I) varied from 1.29 to 1.84. The lowest PI values—indicative of a higher discriminating power—were found for CaT-B504 (PI = 0.05) and CaT-B503 (PI = 0.07), confirming their high efficacy in genotype differentiation.

Both observed (Ho) and expected (He) heterozygosity values were relatively high across the analyzed loci (Table 1), further indicating a substantial level of genetic variation within this wild *С. avellana* population.

### 3.3. Structure and Phylogenetic Analysis

A total of 105 hazelnut cultivars were included as a reference group to compare with the wild *C. avellana* accessions in the STRUCTURE analysis. The most probable number of genetic clusters (K) was determined using the ΔK method, which identified the highest ΔK value at K = 3. The corresponding population structure is illustrated in Figure 3.

The inferred structure at K = 3 suggests the presence of three distinct genetic groups. All 60 wild accessions from Western Kazakhstan were assigned to a single cluster (cluster 3 in Figure 3), whereas the 105 cultivated accessions were distributed across the remaining two clusters (clusters 1 and 2 in Figure 3). This clear genetic separation highlights the significant divergence between wild and cultivated gene pools.

This pattern was further supported by a principal coordinate analysis (PCoA) performed in GenAlEx, which confirmed the marked genetic differentiation between the wild *C. avellana* accessions from Western Kazakhstan and the cultivated varieties (Figure 4).

The phylogenetic tree constructed using UPGMA cluster analysis divided the 165 *C avellana* genotypes including both wild Kazakhstan samples and cultivars into three main clusters. These findings are consistent with the results obtained by STRUCTURE and PCoA analysis. The genetic distribution largely reflects the geographical origin of the samples, with the exception of 10 cultivars showing unexpected proximity to wild Kazakhstani populations (Figure 5, Appendix A).

Cluster 1 includes three subclusters. The first includes two Spanish cultivars ‘Martorella’ and ‘Grifoll’; the second includes five cultivars, ‘Contorta’, ‘Gasaway’, ‘Meraviglia di Bollwiller’, ‘Heynick’s Zellernuss’, and ‘Pallagrossa’, which originate from different countries but share a common genetic signature. The third groups all 60 Kazakh wild samples and the 2 Italian cultivars ‘Tonda Bianca’ and ‘Tonda Rossa.

Cluster 2 is also divided into three subclusters and groups cultivars from the Southern European–Mediterranean region (Italy, Spain, Portugal, Greece, Turkey, and Macedonia). The first subcluster comprises 14 cultivars from Turkey and Greece. The second subcluster includes 17 samples, while the third contains 41 samples. The last two subclusters are composed of cultivars mainly from Italy and Spain, with the exception of the two American cultivars ‘Royal’ and ‘Willamette’.

Cluster 3 is divided into two subclusters containing 4 and 19 samples and mainly includes cultivars from the United States and the United Kingdom, as well as samples of unknown origin. The Spanish cultivar ‘Casina’ is also grouped within the second subcluster.

Spanish cultivars are present across all three clusters, highlighting their high genetic diversity and reflecting the complex history of hazelnut breeding.

### 3.4. In Vitro Conservation of Corylus avellana L.

Due to the absence of viable seeds (nuts), one-year-old vegetative shoots from 27 accessions of *C. avellana* were collected in November 2023 for the initiation of in vitro cultures. The cuttings were initially placed in vessels containing distilled water and maintained at room temperature (24 ± 1 °C) to induce bud break and shoot development. After 1–2 weeks, actively growing apical shoots measuring 2–3 cm in length were excised and used for in vitro culture initiation (Figure 6).

Initial disinfection attempts using standard protocols with chlorine-based agents (commercial bleach) and mercuric chloride proved ineffective due to the high level of contamination of the explants. Over 90% of shoot apices treated with conventional sterilization procedures exhibited contamination by fungi and bacteria. To address this issue, an additional pre-treatment step using 5% (*v*/*v*) Plant Preservative Mixture (PPM) with gentle shaking was introduced, which significantly improved explant sanitation. A comparative analysis of disinfection durations showed that 30 min of treatment with 5% PPM resulted in a fungal contamination rate of 82.5% and a low explant viability of 11.1% (Figure 7). Extending the PPM exposure to 60 min substantially reduced contamination to 44.4% and increased the survival rate of green viable shoots to 48.1%. However, 90 min of treatment led to a marked increase in necrosis (41.9%), reducing the number of viable explants to 18.6%.

To assess latent (endophytic) contamination, visually healthy shoots were transferred to a diagnostic medium (523 detection medium [43]). Among 54 cultured explants, fungal contamination was detected in 14.8% (Figure 7) and bacterial contamination in 5.6% (Figure 7), while 79.6% (*n* = 43) remained aseptic. Contaminated shoots were discarded and only aseptic shoots were selected for subsequent micropropagation (Figure 6C).

As a result of the optimized disinfection and culture initiation procedures, 19 out of 42 collected one-year-old shoots (45.2%) were successfully established under in vitro conditions. A two-step disinfection protocol was developed for highly contaminated material: preliminary immersion in 5% (*v*/*v*) PPM solution for 60 min, followed by treatment with 0.1% mercuric chloride for 7 min. For culture initiation and multiplication, explants were placed on Hazelnut Medium as described by Akin et al. [42], supplemented with 4.4 μM BAP, 0.04 μM IBA, and 20 g·L^−1^ glucose, with pH adjusted to 5.5. Under these conditions, the multiplication rate ranged from 2.8 to 4.0.

In vitro collection of aseptic *C. avellana* plantlets from the Kazakhstan population was successfully established, providing a valuable platform for further conservation and propagation of this rare and endangered genetic resource.

In addition to ex situ approaches, the conservation of *C. avellana* in Kazakhstan should include strict protection of the existing population in the Dubrava reserve, habitat management to maintain forest structure and hydrological balance, prevention of logging or grazing, and regular monitoring and community engagement to reduce anthropogenic pressure.

## 4. Discussion

The results of this study indicate further deterioration in the condition of the only known population of *Corylus avellana* in Kazakhstan compared to previous assessments. In 2018, approximately 30% of the shrubs bore sparse fruits [6]; however, field expeditions conducted in 2023 and 2024 revealed complete absence of fruiting. Additionally, prolonged spring flooding contributed to plant decay, further exacerbating the already critical state of the population.

Molecular genetic analysis using ten microsatellite markers revealed a moderate level of genetic diversity within the Kazakh population. The average number of alleles per locus was 12, comparable to findings reported by Öztürk et al. [17], who reported an average of 10.3 alleles per locus in hazelnut populations. The observed polymorphism information content (PIC) in this study was 0.70, consistent with the values reported by Martins et al. [45], who found a mean PIC of 0.778. These results confirm the informativeness and discriminatory power of the SSR markers used and support previous observations that hazelnut populations, including wild accessions, are highly heterozygous [46,47].

Population structure analysis using STRUCTURE 2.3.4. software revealed clear genetic differentiation between the Kazakh wild population and European cultivated varieties. Similar results were obtained by Martins et al. [19], who demonstrated the separation of landraces, wild accessions, and cultivated forms into distinct genetic clusters, although using AFLP and ISSR markers. This genetic distinctiveness of the Kazakh population highlights its potential importance as a reservoir of unique alleles and adaptive traits, reinforcing its priority status for conservation.

A phylogenetic tree was constructed based on the analysis of 10 SSR loci, including both cultivated *C. avellana* varieties and wild populations collected in Kazakhstan. The resulting tree showed the presence of three main clusters based on genetic background. Previous studies on hazelnut have similarly assigned cultivars within phylogenetic trees according to their genetic background [13,21,48,49].

The first cluster includes two Italian cultivars, ‘Tonda bianca’ and ‘Tonda rossa’, together with the Spanish cultivars ‘Martorella’ and ‘Grifoll’, as well as Kazakh wild populations. This is consistent with the hypothesis that Italian and Balkan populations represent glacial refugia that have preserved an ancient gene pool [50]. Genetic studies further demonstrate the unique structure of local varieties, which differs from later fruit lines [51]. Regional Spanish cultivars, such as ‘Martorella’ and ‘Grifoll’, also show distinct genetic profiles, confirming their special origin and proximity to the wild forms. The third subcluster also includes six cultivars of different origins, which formed a distant cluster in the study by Boccacci et al. [28].

The second cluster groups cultivars from Italy, Spain, Portugal, Greece, and Turkey, with genotype distribution of s generally reflecting their geographical origin. An exception was observed with the American cultivars ‘Royal’ and ‘Willamette’, which clustered close to the European genotypes. This pattern likely reflects their European origins, consistent with previous data showing that American hazelnut varieties are genetically close to their original European lines [13].

The third cluster comprises the Spanish cultivar ‘Casina’, along with varieties from the United States and the United Kingdom and some samples of uncertain origin. The cultivar ‘Casina’ was notably distinct from the main group of European cultivars in the phylogenetic tree. This observation aligns with findings from Ferreira et al. [52], in which ‘Casina’ also formed a separate branch, showing marked distance from other Spanish and European genotypes based on morphological traits and ISSR markers. Furthermore, chloroplast microsatellite analysis by Boccacci & Botta [53] showed that several traditional varieties, including the Spanish ones, have rare and unique chlorotypes. This suggests that ‘Casina’ may preserve ancient genetic lineages not commonly found in modern European cultivars.

Given the ongoing degradation of the natural population, the preservation of *C*. *avellana* in Kazakhstan through in situ methods alone appears insufficient. Environmental stressors, such as seasonal flooding, combined with the lack of seed regeneration and biotic damage, pose a significant threat to the long-term survival of this relict population. In this context, the application of modern biotechnological approaches—particularly in vitro micropropagation—offers a viable alternative for the conservation and restoration of this endangered genetic resource.

Initiating in vitro cultures from woody perennials such as *C. avellana* presents considerable challenges, primarily due to the high levels of epiphytic and endophytic contamination. In the present study, the collected vegetative material exhibited a high incidence of epiphytic fungal contamination as well as endophytic bacterial presence. Similar challenges have been reported in other studies on hazelnut micropropagation, where it was shown that shoots used as explants were colonized by a wide variety of bacterial pathogens belonging to various genera: *Pseudomonas*, *Brevundimonas*, *Agrobacterium*, *Xanthomonas*, *Enterobacter*, and others [54,55]. To mitigate contamination, researchers have recommended the use of antibiotics [54] or thermotherapy of donor plants [56]. Improved disinfection protocols for woody plant explants with additional PPM treatment have resulted in significant reductions in contamination and significantly increased yields of aseptic shoots in species such as walnut [57], rhododendron and European birch [58], and acacia [59]. Similarly, in vitro incubation of apple and blackberry shoots with identified endophytic contamination on medium supplemented with 0.2% PPM has proven effective for controlling endophytic bacteria growth and maintaining aseptic shoots in in vitro collections [60,61]. The findings of the present study demonstrate that incorporating a pre-treatment step using the broad-spectrum biocide PPM significantly reduced the contamination level from over 90% to 44.4%, while increasing the percentage of viable green explants to 48.1%.

## 5. Conclusions

The results of this study demonstrate that the sole natural population of *Corylus avellana* L. in Kazakhstan is critically endangered due to its limited size, the absence of generative reproduction, and exposure to ongoing environmental stressors. Despite its small geographic extent, the population maintains high levels of nuclear genetic diversity, highlighting its importance as a unique genetic resource. Nuclear markers confirm its distinctiveness. The successful establishment of an in vitro collection represents a significant step toward ex situ conservation and provides a practical tool for future propagation and restoration efforts. To secure the long-term survival of this relict population, integrated measures are required, combining in situ protection, strict habitat management, and regular monitoring. In addition, the authors propose the establishment of a field-based mother plantation (clonal orchard) derived from both natural and in vitro-propagated material. This plantation will function as a living gene bank and a source of planting stock for future reinforcement and restoration programs, representing a concrete step toward the practical implementation of this study’s results.

## Figures and Tables

**Figure 1 biology-14-01472-f001:**
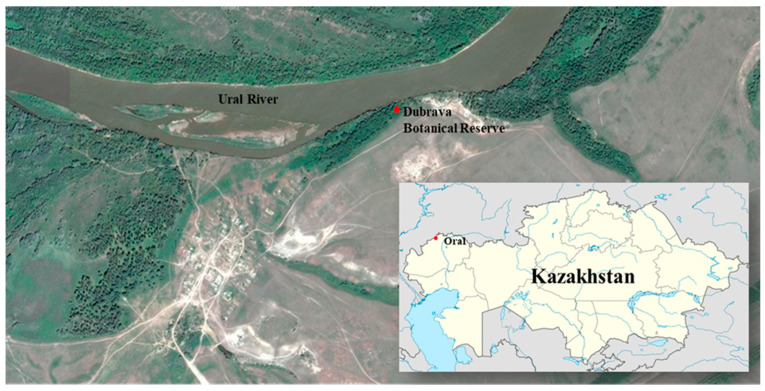
The collection site of *Corylus avellana* L. in the Western Kazakhstan region (marked with red dots).

**Figure 2 biology-14-01472-f002:**
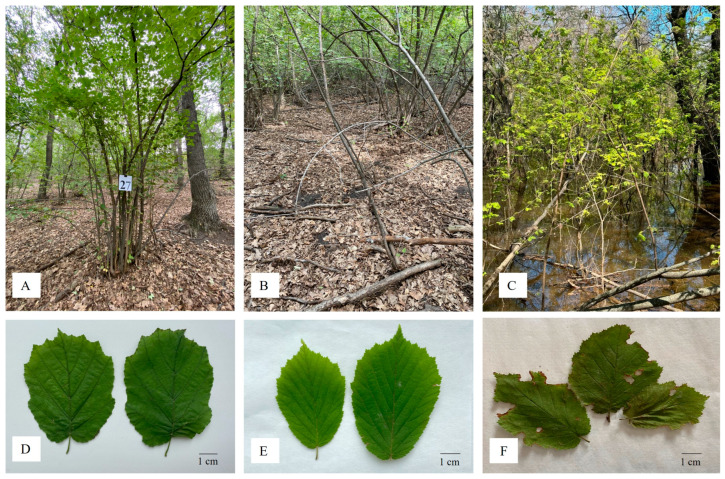
The condition of the *Corylus avellana* L. plants in the wild population: (**A**) mature; (**B**) dying; (**C**) flooded in May 2024. Leaf condition and shape: (**D**) rounded; (**E**) ovate; and (**F**) leaves affected by pests (scale bar 1 cm).

**Figure 3 biology-14-01472-f003:**
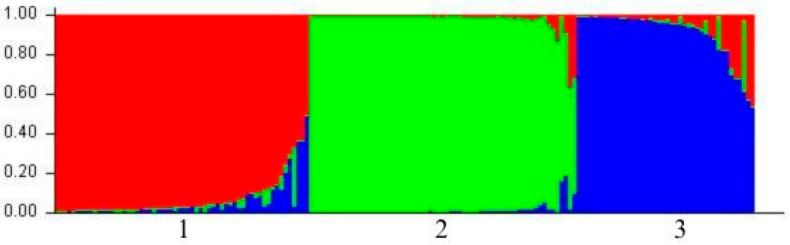
Genetic structure analysis of 60 samples of wild *Corylus avellana* L. and 105 cultivars. Population structure inference by Bayesian assignment with K = 3. Each individual is represented by a vertical line. The colors indicate the proportional membership of each genotype in the inferred clusters (cluster 1 = red, cluster 2 = green, cluster 3 = blue). Cluster 1 comprises cultivated accessions originating from Turkey, Greece, and Spain. Cluster 2 comprises cultivars from Spain, Italy, the United States, and the United Kingdom. Cluster 3 groups all 60 wild *C. avellana* accessions collected in Kazakhstan.

**Figure 4 biology-14-01472-f004:**
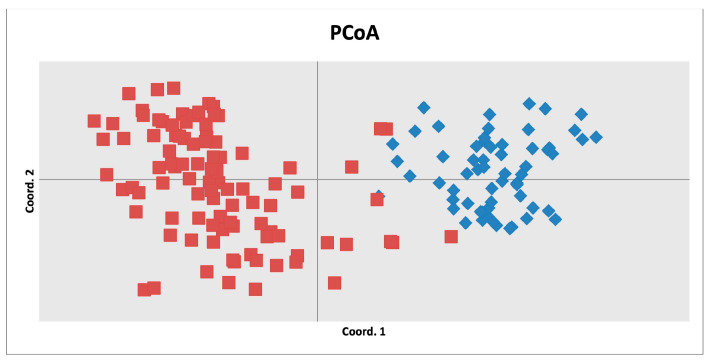
Principal coordinate analysis (PCoA) of pairwise distance among hazelnut cultivars (in red) and wild *C. avellana* (in blue).

**Figure 5 biology-14-01472-f005:**
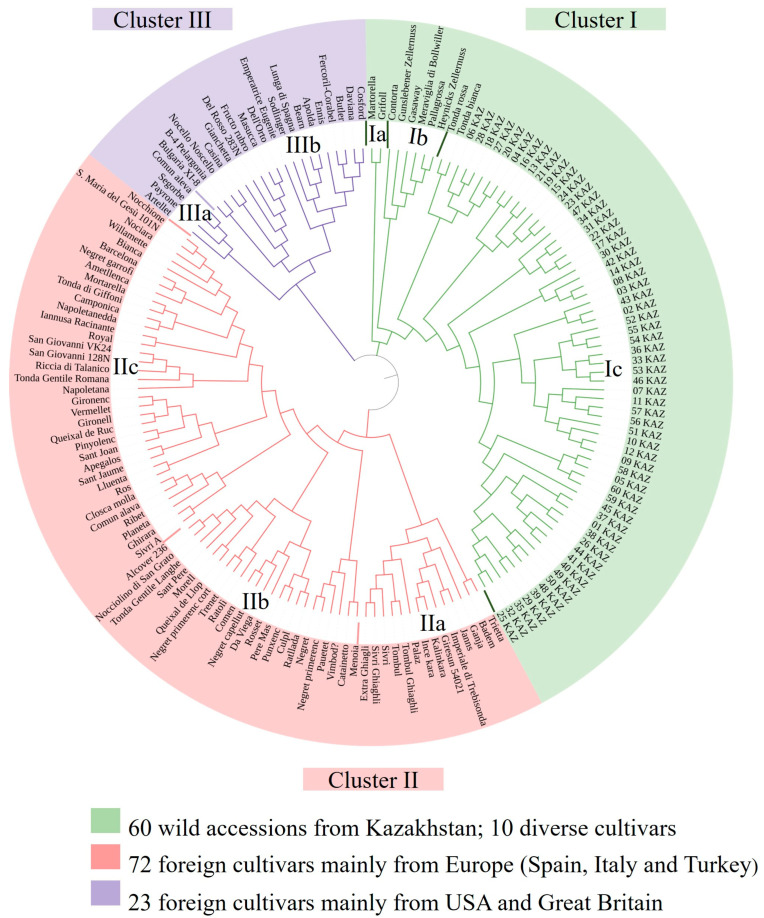
A phylogenetic tree of 165 cultivated and wild *C. avellana* accessions, including Kazakhstani populations, constructed using the UPGMA method. The three main clusters are indicated by different colors and subclusters are labeled with Roman numerals within the phylogenetic tree.

**Figure 6 biology-14-01472-f006:**
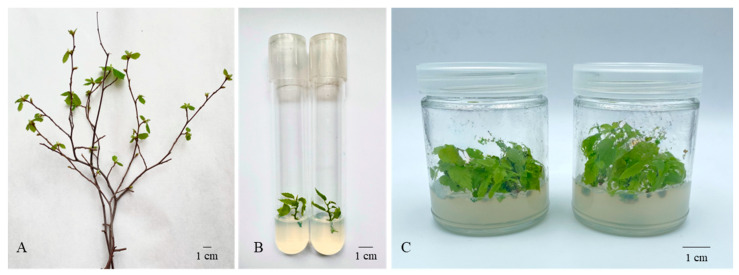
*Corylus avellana* L. in vitro culture initiation. (**A**) Sprouted shoots used as explants; (**B**) shoot apex on Akin et al.’s [42] Hazelnut Medium with 4.4 μM BAP, 0.04 μM IBA, 20 g·L^−1^ glucose, 4 g·L^−1^ agar, and 1.75 g·L^−1^ Gerlite™, pH 5.5. (**C**) Micropropagated aseptic shoots of *Corylus avellana* (scale bar: 1 cm).

**Figure 7 biology-14-01472-f007:**
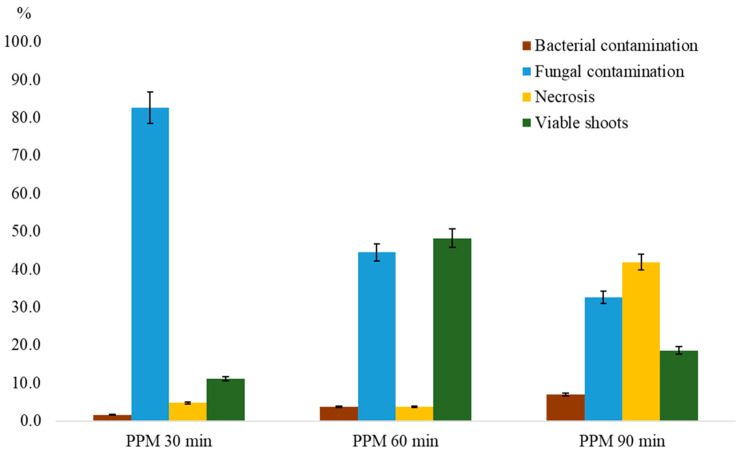
The effect of treatment duration with 5% (*v*/*v*) Plant Preservative MixtureTM (PPM) on in vitro culture initiation of *Corylus avellana* L. Data are presented as the mean value ± standard error (M ± SE). Means with different letters differ significantly at *p* < 0.05.

**Table 1 biology-14-01472-t001:** Summary statistics for 60 wild *Coryllus avellana* L. accessions over 10 SSR loci.

	CaT-B505	CaT- B503	CaT- B507	CaC- B020	CaT- B501	CaT- B107	CaT- B504	CaT- B502	CaC- B028	CaT- B508	Mean	SE
Na	9	12	10	14	10	16	12	12	13	12	12	-
Ne	3.09	5.02	4.14	3.06	3.81	3.46	5.84	3.68	2.9	3.61	3.86	0.09
Ho	0.70	0.85	0.81	0.57	0.76	0.78	0.87	0.68	0.46	0.52	0.7	0.04
He	0.68	0.80	0.76	0.67	0.74	0.71	0.83	0.73	0.65	0.72	0.73	0.02
PIC	0.62	0.77	0.72	0.62	0.71	0.66	0.81	0.7	0.61	0.69	0.7	-
PI	0.16	0.07	0.1	0.16	0.1	0.13	0.05	0.1	0.17	0.11	0.26	-
I	1.3	1.68	1.51	1.35	1.64	1.51	1.84	1.59	1.29	1.56	1.53	0.05

Na: number of alleles; Ne: mean effective number of alleles per locus; Ho: observed heterozygosity; He: expected heterozygosity; PIC: polymorphism information content; PI: probability of identity; I: Shannon–Weaver Diversity Index; SE: standard error.

## Data Availability

The presented research results are included in the article and Appendix A. Further inquiries can be directed to the corresponding author.

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
