# Peer review of "Genetic Diversity of the Only Natural Population of Corylus avellana L. in Kazakhstan and Prospects for Its In Vitro Conservation"

_biology, 2025, doi:10.3390/biology14111472_

Round 1

Reviewer 1 Report

Comments and Suggestions for Authors

This manuscript makes a valuable contribution to plant conservation genetics and it also demonstrates practical conservation action. The research addresses a clear conservation need for valuable genetic material with appropriate methodology. The combination of field surveys, genetic analysis, and practical conservation action (in vitro cultures establishment) represents a comprehensive approach. The genetic findings provide valuable insights into the population's distinctiveness and conservation value.

The study design is appropriate, combining field surveys with robust molecular analysis using ten nuclear and ten chloroplast SSR markers. An establishment of an in vitro collections in laboratory conditions represents a practical conservation achievement. The authors demonstrate clear expertise in both molecular genetics and tissue culture techniques.

I have several suggestions for manuscript improvement:

Methodology

Line 124-125: Consider providing more detail on the silica gel drying protocol duration and storage conditions before DNA extraction. For example: how long the samples were with silica gel.

Lines 165-179: The in vitro culture initiation protocol could benefit from mentioning several details: type of light source (cool white or LED?) and manufacturer of light sources for in vitro cultivation; further more precise description of time of treatment with 0.1% mercury chloride,

more detailed description of in vitro media autoclaving procedure.

Results and discussion

Section 3.1: While the phenotypic description is adequate, consider including a brief descriptioon on possible pest damage severity or specific disease symptoms observed, which could add valuable information for future management strategies.

If the population had been healthy, without visible systemic infections, I would state this fact explicitly."

Figure 3: Scale bars are mentioned in explanation but not described on the photographs of leaves. I would recommend all scale bars are clearly marked with numbers directly on the photographs.

The authors could strengthen the conservation section by discussing specific recommendations for in situ habitat management or protective measures in the locality.

Line 205-207: The phrase "in clustered formations with regeneration occurring exclusively through vegetative shoot formation" could be clarified to specify clonal vs. sexual reproduction patterns.

Lines 447-457: The conclusions effectively summarize the key findings but could be more specific about the recommended conservation actions. Conclusion should also mention concrete further plans/actions of the author team aimed at further implementation and exploitation of study/article findings and results.

Author Response

Thank you very much for taking the time to review this manuscript. Please find the detailed responses below and the corresponding revisions/corrections highlighted/in track changes in the resubmitted files. Please see the attachment.

Reviewer 2 Report

Comments and Suggestions for Authors
  • Authors  evaluated the morphological and genetic
    diversity of a relict C. avellana population in Kazakhstan using SSR markers, and established an in vitro culture of this population. The evaluation of genetic diversity of an endagered population reflect the will to design conservation strategies with impact in local ecology.
  • Comments
  • the period of lines 96-97 "Evaluating the genetic diversity of C. avellana is critical for breeding programs, conservation of valuable alleles, and for understanding responses to biotic and abiotic" should be modified: is not certain that genes from a wild population are critical for breeding although it is true for conservation.
    The hypothesis of diversity of a wild populatiion was proved as well as the establishment of an in vitro culture protocol. Methodology is appropriate. 
  • Chloroplast diversity is deficiently presented in the Results section. The number of haplotypes found and respective identification should be added to the manuscript.
  • Figure 9 legend repeat description of figure 8. Must be replaced.
Comments on the Quality of English Language

English is acceptable but can be improved.

Author Response

Thank you very much for taking the time to review this manuscript. Please find the detailed responses below and the corresponding revisions/corrections highlighted/in track changes in the re-submitted files. Please see the attachment.

Reviewer 3 Report

Comments and Suggestions for Authors
  1. Figures 1 and 2, which serve as collection point information, can be included as supplementary materials and do not need to be placed in the main text. It is recommended to include images of the experimental materials used in this study, such as Figure 3, in the main text instead.

  2. The layout of Table 1 is not aesthetically pleasing and needs adjustment. It should be arranged clearly with horizontal rows and vertical columns for better readability.

  3. Figure 4a lacks substantial content. If it must be included in the text, it is advised to reduce its size.

  4. Similarly, Figure 5 has limited content but a large size. Additionally, why are the legends for cultivated and wild materials different in size? The overall quality of the figure is low and not clear enough.

  5. For Figure 6, the black border can be removed. Clusters I, II, and III should be adjusted so that the arcs closely fit the respective clusters. Furthermore, the labels for each material in the cluster tree are too small and should be enlarged.

  6. The plant materials shown in both subfigures of Figure 7 are not clear enough. It is recommended to replace them with high-resolution and visible images.

  7. For Figure 8, the gridlines on the vertical axis should be removed.

  8. In Figure 9a, the use of arrows to label "abc" is not recommended. Moreover, why are there corresponding labels in the figure but no explanations in the caption? Additionally, why are there lines on both the top and bottom of the black marker on the petri dish? This is not visually appealing. If the top lines are the intended ones, it is suggested to erase the bottom lines and retake the photo.

  9. There are too many figures, and it is recommended to delete or merge some. The actual content is limited, or even insufficient. As mentioned earlier, Figures 1 and 2 can be moved to the supplementary materials. The remaining seven figures should be merged based on content needs. For example, can Figures 4 and 5 be merged? Can Figures 7, 8, and 9 be merged?

  10. Two different citation formats appear in the text. It is recommended to adhere to the journal's guidelines. For example, in line 416: "findings from Ferreira et al. (2010), where Casina also formed a separate branch."

Author Response

(The authors gave the same response as above.)

Round 2

Reviewer 1 Report

Comments and Suggestions for Authors

The authors have revised the manuscript. I agree with the publication of this article.

Reviewer 2 Report

Comments and Suggestions for Authors

Reviewing accepted